# Reconstruction of the unbinding pathways of noncovalent SARS-CoV and SARS-CoV-2 3CLpro inhibitors using unbiased molecular dynamics simulations

**Fereshteh Noroozi Tiyoula⦿, Hassan Aryapour⦿ ***

Department of Biology, Faculty of Science, Golestan University, Gorgan, Iran

⦿ These authors contributed equally to this work.
* hassan.aryapour@gmail.com

**Data Availability Statement:** All data and analysis files are available on Zenodo (DOI: 10.5281/

## Abstract

The main protease (3CLpro) is one of the essential components of the SARS-CoVs viral life cycle, which makes it an interesting target for overpowering these viruses. Although many covalent and noncovalent inhibitors have been designed to inhibit this molecular target, none have gained FDA approval as a drug. Because of the high rate of COVID-19 pandemic development, in addition to laboratory research, we require *in silico* methods to accelerate rational drug design. The unbinding pathways of two SARS-CoV and SARS-CoV-2 3CLpro noncovalent inhibitors with the PDB IDs: 3V3M, 4MDS, 6W63, 5RF7 were explored from a comparative perspective using unbiased molecular dynamics (UMD) simulations. We uncovered common weak points for selected inhibitors that could not interact significantly with a binding pocket at specific residues by all their fragments. So water molecules entered the free binding S regions and weakened protein-inhibitor fundamental interactions gradually. N142, G143, and H163 are the essential residues, which cause key protein-ligand interactions in the binding pocket. We believe that these results will help design new potent inhibitors against SARS-CoV-2.

## Introduction

Severe Acute Respiratory Syndrome (SARS) occurred in Guangdong Province of China in 2002–2003, which was caused by SARS-CoV-1, a coronavirus of 2b β-coronavirus [1]. A novel coronavirus (2019-nCoV) was identified in Wuhan, China, in late 2019 for the first time [2]. This virus, which was scientifically named severe acute respiratory syndrome coronavirus 2 (SARS-CoV-2), has infected many people in different parts of the world with its high prevalence power and caused the COVID-19 pandemic with symptoms including fever, cough, and fatigue [3, 4]. Sadly, now after two years, according to the World Health Organization(WHO) reports, 4,777,503 people have lost their lives all around the world (Updated on October 03, 2021) [5].

zenodo.4587009). Also, in the Methods section, all used software is listed.

**Funding:** This investigation was supported by grant number 99-213-1 from Golestan University, Gorgan, Iran. The funders had no role in study design, data collection and analysis, decision to publish, or preparation of the manuscript.

**Competing interests:** The authors have declared that no competing interests exist.

Coronaviruses are single-stranded positive-sense RNA viruses with the largest genome, approximately 30 kilobases, among all known RNA viruses. In all Coronaviruses, the genome expression is encoded by the open reading frame (ORF) 1a/b at the 5' end of the genome [6]. Studies have shown that SARS-CoV-2 genes possess about 80% nucleotide identity and 89.10% nucleotide similarity with SARS-CoV genes [7]. Among CoVs, the viral genome of SARS-CoV-2 is about 29.8 kilobase, which encodes two polyproteins that are responsible for viral replication and transcription by proteolytic processing. This virus needs two cysteine pro-teases for these processes, papain-like protease (PLpro) and main proteinase (Mpro), which is also called 3C-like proteinase (3CLpro). Based on pairwise sequence alignment, 96.08% and 98.7% identity and similarity were observed between 3CLpro of SARS-CoV and SARS-CoV-2, respectively [8]. The main protease is a dimer protein with 306 residues and three domains. The domains I and II have antiparallel β barrel structures responsible for the catalytic reaction, while domain III has α-helices and regulates dimerization of the 3CLpro (Fig 1A and 1B). Since only the dimeric form of Mpro is catalytically active, intermolecular interactions between the helical domains play an essential role in activating the enzyme [9, 10].

So, this protein is essential for the viral life cycle and is an interesting target for designing SARS-preventing drugs. To this end, various research groups have been working worldwide, and various approaches were used, including drug repurposing, structure-based design, and fragment-based design [11]. Based on the substrate specificity of 3CLpro, peptidomimetic inhibitors were designed as the first protease inhibitors generation [12]. These inhibitors con-tain Michael acceptors, aldehydes, epoxy-ketones, halo-methyl, and trifluoromethyl ketones, and they form a covalent bond with the catalytic Cys145. In the following, these inhibitors shed light on the idea for further inhibitor design, like nonpeptidic inhibitors with different micromolar ranges [13, 14]. For the SARS-CoV 3CLpro, there are some covalent and noncova-lent inhibitors. At first glance, covalent warheads may seem a priority for overpowering this cysteine protease, but toxicity is one of the major challenges for the therapeutic use of these inhibitors [15]. For this purpose, we focused on reversible noncovalent inhibitors, selected four different inhibitors according to their structures and biological activity. Two of the selected compounds are SARS-CoV 3CLpro inhibitors (PDB IDs: 3V3M and 4MDS), and the other two are SARS-CoV-2 3CLpro inhibitors (PDB IDs: 6W63 and 5RF7) (Fig 2A–2D).

Unfortunately, none of the designed inhibitors has been approved by the FDA. So it is pretty clear that along with experimental researches, *in silico* methods like unbiased molecular dynam-ics (UMD) are essential [16]. MD simulation has been used in many science fields since the 1950s to predict hidden information that cannot be reached through experimental research [17]. The UMD method eliminates artificial interactions between protein and inhibitor because it does not apply any biasing forces or potentials to the simulations [18]. Studying the unbinding mechanisms of inhibitors in complex with their target proteins is one of the great features of the UMD simulation method. In judging candidate drugs, the bioavailability, selectivity, metabolic properties, and binding affinity of the designed inhibitor to its target protein are important [19]. In addition to these parameters, the mean lifetime that the drug remains in the binding site is equally important. Experimental techniques can measure the time it takes for a drug to unbind from a target, but the essence of the matter is much deeper than a number [20]. On the other hand, by investigating the unbinding pathways of particular inhibitors, important information involving: protein-ligand key interactions, ligands interacting efficiently with the target can be obtained. Finally, a fully atomistic scenario will be presented based on the obtained results [21]. As a result, many research groups have examined unbinding pathways of various drugs or inhibitors over the years via MD simulation methods and prepared a solid foundation for ratio-nal drug design. [22, 23]. Among advanced MD approaches, the supervised molecular dynamics (SuMD) [24, 25] method is relatively novel. With a tabu-like supervision algorithm, it is possible

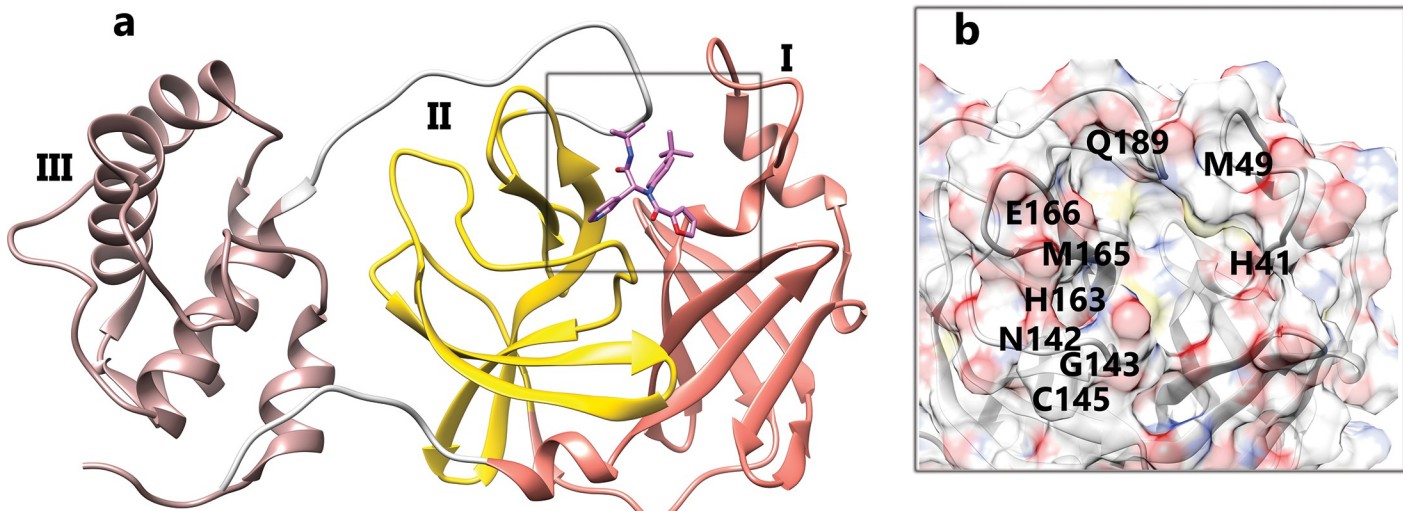

**Fig 1.** The 3D ribbon structure of the 3CLpro (PDB ID 3V3M) complexed with ML188 **A**, The protein domains: Domain I (r8-r101), Domain II (r102-r184), Domain III (r201-r306). **B**, The important active site residues.

to fully unbind small molecules from their molecular targets within very short times without applying any biasing force or potential. This method also produces information regarding meta-stable intermediate ligand-bound states, which are essential for rational drug design. In this regard, the SuMD was used to examine different cases of protein-ligand recognition mechanism, involving: the human casein kinase 2 (CK2) complexed with ellagic acid, the P1-1 isoform of glutathione S-transferase (GSTP1-1) complexed with sulfasalazine, the human peroxiredoxin 5 (PRDX5) complexed with benzene-1,2-diol, and the human serum albumin (HSA) in complex with (S)-naproxen [25].

Half-maximal inhibitory concentration ($IC_{50}$) of SARS-CoV 3CLpro inhibitors are measured experimentally before, and the 3CLpro of SARS-CoV and SARS-CoV-2 are approximately the same, without any difference in the binding site [14, 26]. Therefore, based on participation in the solidarity clinical trial of COVID-19 treatments, our research team decided to compare the noncovalent SARS-CoV and SARS-CoV-2 protease inhibitors' unbinding pathways using the SuMD.

## Methods

All simulations were originated from X-ray crystallography of 3CLpro-ligand complex in Protein Data Bank (PDB IDs: 3V3M [14], 4MDS [26], 6W63 [27], 5RF7 [28]). At first, missing atoms and residues of proteins were added and fixed using UCSF Chimera software [29]. Then ligands were parameterized by ACEPYPE using default settings (the GAFF atom type and BCC partial charges) [30]. After preparation, protein-ligand complexes were constructed in GROMACS 2018 [31] using AMBER99SB force field [32] and TIP3P water model [33]. Selected holo-proteins were located in the center of triclinic boxes with a distance of 1.2 nm from each edge. The next step was to provide a 150 mM neutral physiological salt concentration, sodium, and chloride ions. Then all systems were relaxed in energy minimization using the steepest descent algorithm and reached Fmax of less than 1000 kJ.mol$^{-1}$.nm$^{-1}$. Using the Linear Constraint Solver (LINCS) algorithm, all Covalent bonds were constrained to maintain constant bond lengths [34]. The long-range electrostatic interactions were treated using the Particle Mesh Ewald (PME) method [35], and the cut-off radii for Coulomb and Van der

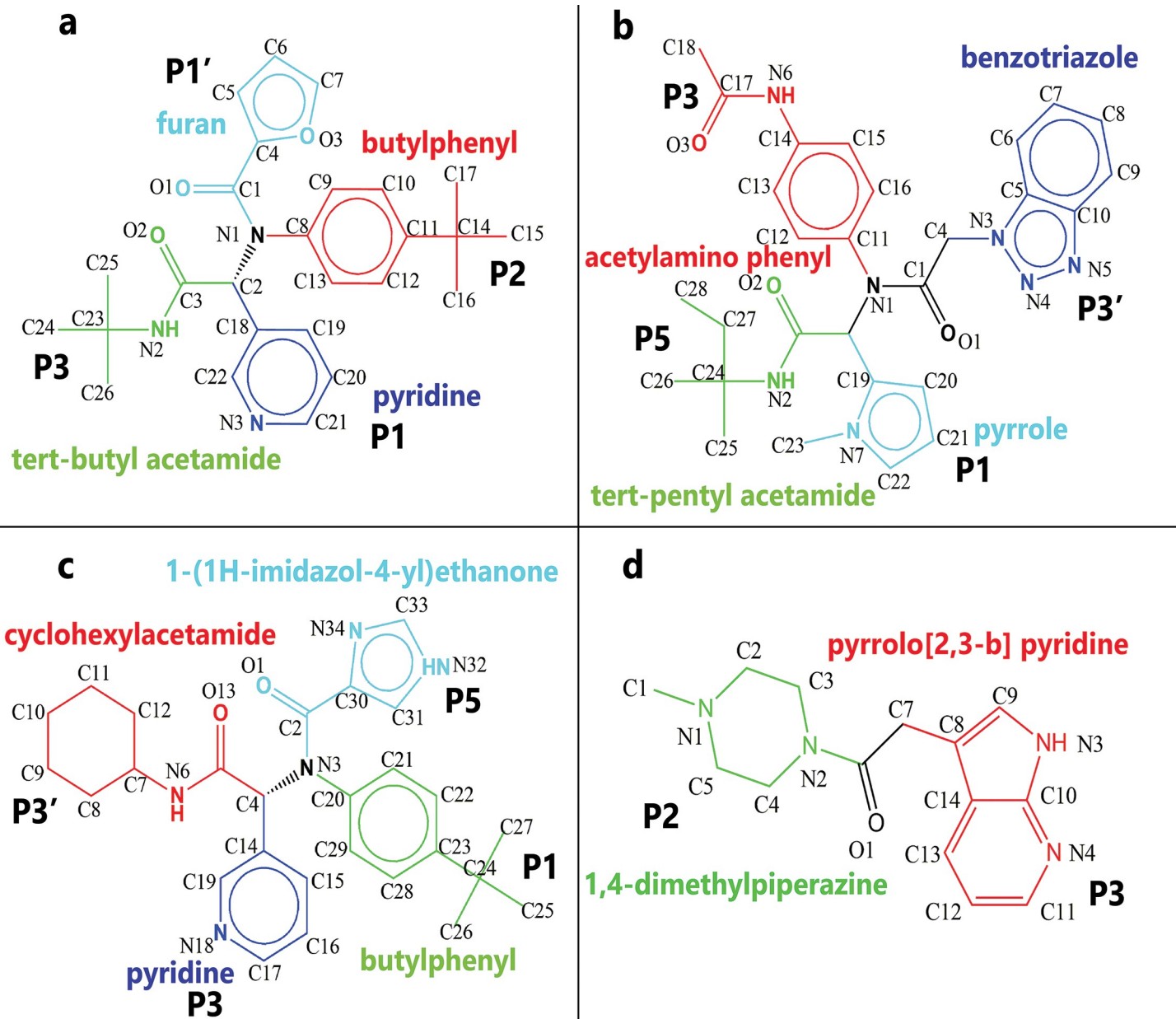

**Fig 2. The fragmented form of the 2D structure of selected inhibitors and the. A**, The structure of ML188 in PDB ID: 3V3M and occupied positions of binding pocket by inhibitor are represented by "P" letters. **B**, The structure of ML300 in PDB ID: 4MDS and occupied positions of binding pocket by inhibitor are represented by "P" letters. **C**, The structure of inhibitor3 in PDB ID: 6W63 and occupied positions of binding pocket by inhibitor are represented by "P" letters. **D**, The structure of inhibitor4 in PDB ID: 5RF7 and occupied positions of binding pocket by inhibitor are represented by "P" letters.

Waals (VdW) short-range interactions were set to 0.9 nm for all systems. Finally, the modified Berendsen (V-rescale) thermostat [36] and Parrinello-Rahman barostat [37] were applied for 100 and 300 ps for the equilibrations and keep the system in stable environmental conditions (310 K, 1 Bar) and got ready to begin molecular dynamic simulations with a time step of 2 fs and without applying any human or non-human biasing force or potential. In this regard, to reach complete unbinds, we performed 12 separate series of replicas (three replicas for each complex), with fixed duration times by the SuMD method with some modifications. Herein,

we set the center of mass (COM) of ligands as a first spot, and the COM of His41, Cys145, His163, Asp187 in PDB ID 3V3M, His41, Met49, Cys145, His164 in PDB ID 4MDS, and His41, Cys145, His163, Met165, Gln189 in PDB IDs 6W63 and 5RF7 as second spots and, ran all simulations with a time window of 500 ps. After finishing each run, the frame with the longest distance between selected spots was selected automatically to extend the next 500 ps simulation. These processes were continued until complete unbinding was obtained, which is equal to a distance of 50 Å between the mentioned spots. Finally, all events in every concatenated trajectory file were investigated carefully with GROMACS utilities for data analysis. Figures were created using UCSF Chimera and Daniel's XL Toolbox (v7.3.4) [38]. In addition, Matplotlib was used to create the free energy landscape plots to visualize the essential interactions [39]. The free energy landscapes plots were made based on three variables time, ligand RMSD, and protein RMSD. The ligand and protein RMSD values were selected because they were meaning full and had sharp changes as a function of time during unbindings. Analyzing these plots can reveal the stable states of inhibitors, as well as the residence time of inhibitors in each state over unbinding. Areas that tend to turn blue color indicate that the inhibitor has been present in this area for a longer time.

## Results and discussion

One of the selected compounds, ML188 (PDB ID 3V3M) with an $IC_{50}$ of 4.11 μM [14], were simulated in 3 replicas, and the complete unbinding processes occurred at the times of 60, 40, and 37 ns (Fig 3A). In the first and longest replica, during the first state (Fig 3K), the ligand was enclosed within the binding pocket VdW forces for 55 ns (Fig 3G). However, all residues were not equally important; the most prominent VdW interactions formed between Met49 and Met165 residues and butylphenyl fragment of the inhibitor in its binding pose (Fig 3B). By rotation of the furan ring in inhibitor, pyridine fragment formed the third prominent VdW and amino-pi interactions with Gln189 (Fig 3C and 3G). Presumably, this rotation occurred because, unlike the butylphenyl fragment, which was well held by two methionines, other fragments did not significantly interact with the binding pocket at a specific residue. So the ligand moved from the deeper part of the binding pocket toward the exit area. These interactions became weaker due to the formation of H-bonds between the oxygen atoms in the furan ring and Ala46 and Glu47's backbones. (Fig 3F and 3H). Since these H-bonds pulled the ligand out of the catalytic site completely (Fig 3D), the last protein-ligand interactions in the second short intermediate state (~ 10 ns) cannot be considered as essential bonds because they formed out of binding pocket and just increased simulation time (S1 Video).

In the two other replicas (rep2 and 3), the inhibitor unbound completely sooner due to the lack of H-bond formation with Ala46 and Glu47, but in comparison with the rep1, in addition to those three essential residues, His41 with VdW interactions with butylphenyl group of ligand was the fourth prominent residue (Fig 3E, 3I and 3J). The His41, due to its good position in the binding pocket, could play a vital keeping role in their single states (Fig 3L and 3M). However, because this residue had pi-sulfur interaction with Cys145, it sometimes loosed its effect on butylphenyl. So the inhibitor among the competition with Cys145 gradually pulled toward Met49 and Gln189 with the help of furan fragment rotations (S2 Video). In laboratory research on this inhibitor, Gly143, Cys145, and His163 were the most important inhibitor-protein interactions in the binding pose. Compared with our results during three replicas, Gly143 and His163 were less important than other critical residues, but Cys145 was important in the rep3 [14].

For this compound, due to the furan-free fragment, water molecules entered into a deep part of the binding pocket in the simulations. (Fig 4A and 4B). These molecules promoted

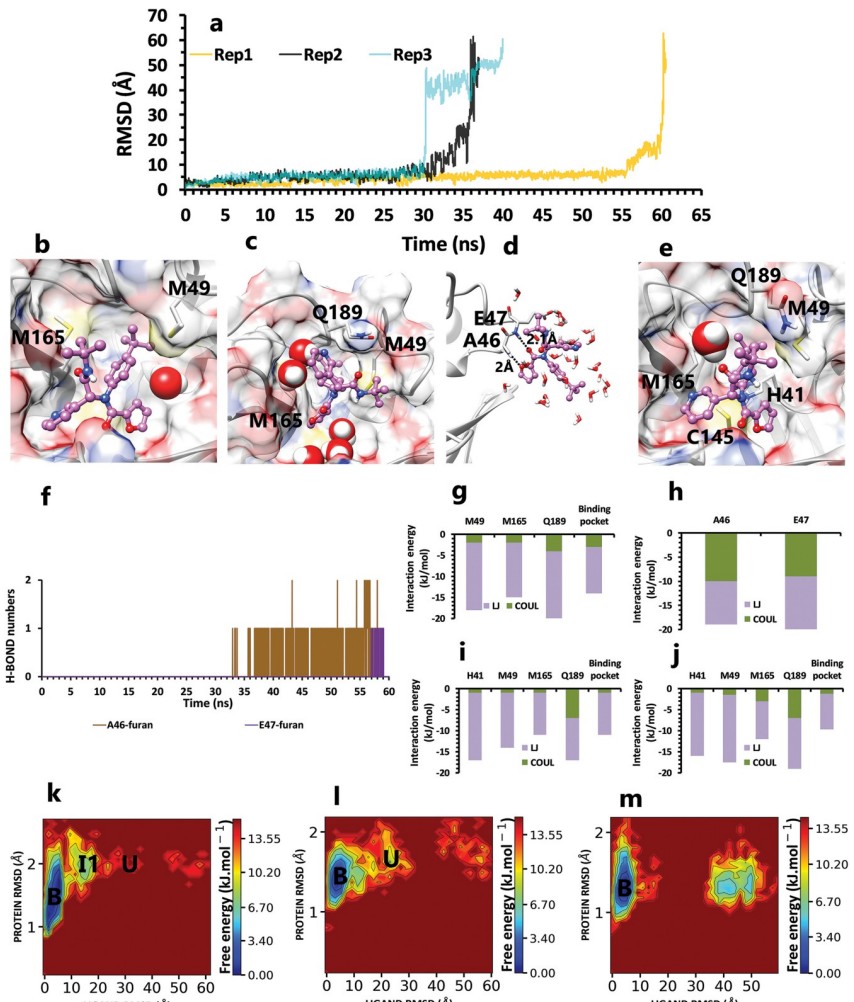

**Fig 3. The details of ML188 unbinding pathways in three replicas. A,** RMSD values of the ligand from binding pose to complete unbinding in three replicas. **B,** The interactions between particular ligand fragments and essential residues in the crystallographic binding pose of rep1. **C,** The new interactions between the inhibitor and binding pocket residues after rotation of the furan ring in rep1 (frame 3030 in the trajectory file). **D,** The last protein-inhibitor interactions before complete unbinding in rep1(frame 5804 in the trajectory file). **E,** The competition between C145 and butylphenyl in interaction with His41 in rep 2 and 3 (frame 27 in trajectory file of rep2). **F,** Hydrogen bond numbers of Ala 46, Glu47 with furan fragment in the second intermediate state of rep1. **G** and **H,** The average of most important interaction energies of the protein-ligand complex in the first and second intermediate state of rep1, respectively. **I** and **J,** The average of most important interaction energies of the protein-ligand complex in rep2 and 3, respectively. **K**, **L** and **M,** The free energy landscape of rep1, 2, and 3 to capture lowest energy stable states of ligand during the unbinding process (bound state (B), intermediate state (I), unbound (U)), respectively, which was calculated using "gmx sham".

unbinding by gradually weakening noncovalent interactions between the inhibitor and protein, thereby allowing a complete unbinding. (Fig 4C).

In the following, the inhibitor, ML300 (PDB ID 4MDS), was selected as a second micromolar noncovalent inhibitor with an $IC_{50}$ of 6.2 μM [26]. In the single state of the first replica in 22 ns (Fig 5A and 5L), pyrrole fragment of inhibitor was in VdW and pi-sulfur interactions with Met49, Met165, and amino-pi and VdW interactions with Gln189 (Fig 5B). Also, the benzotriazole fragment had VdW interactions with Asn142 (Fig 5H), and the $O_1$ atom close to the benzotriazole fragment had H-bond interaction with the backbone of Glu166 (Fig 5G). Except

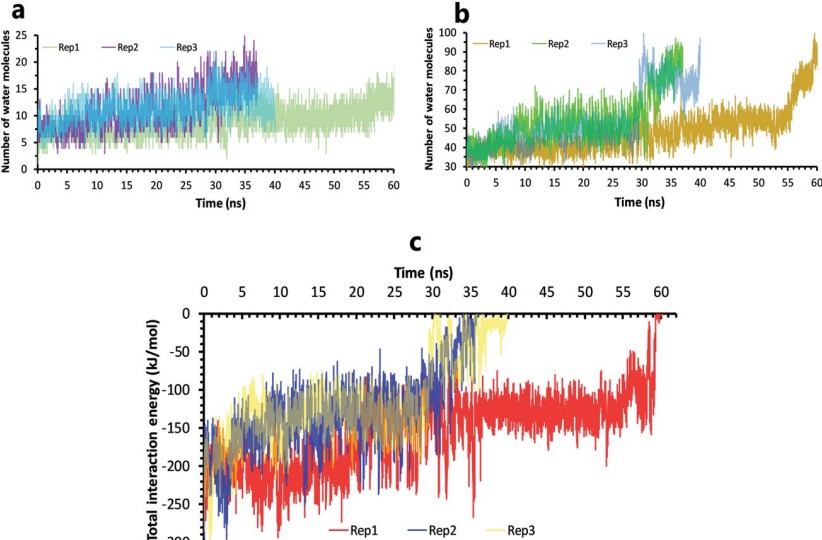

**Fig 4. The details of solvation effects on ML188 unbinding mechanisms. A**, Number of water molecules in the cut-off of 3.5 Å of the binding pocket residues, rep1, 2, and 3. **B,** Number of water molecules in the cut-off of 5 Å of the inhibitor, rep1, 2, and 3. **C,** The total interactions energies of protein-inhibitor complexes in rep1, 2, and 3.

for these two fragments, the other two were not in serious keeping interactions. So by time passing, acetylamino phenyl fragment, due to its close position to Asn142, entered in the competition with benzotriazole fragment and destroyed the effect of Asn142 on benzotriazole (Fig 5C). The most potent inhibitor interaction with the binding pocket was hydrogen bond with Glu166, so by weakening the VdW network (Fig 5F), the ligand unbound from the Glu166 side (S3 Video).

In the other pathway (rep2), by the time of 24 ns, pyrrole fragment had VdW and pi-sulfur interactions with Met49 and Met165, and in contrast with rep1, Gln189 did not have a significant effect (Fig 5J) in the single state of this replica (Fig 5M). Also, the acetylamino phenyl group was not free to break down the interaction of the benzotriazole and Asn142 because it had VdW and cation-pi interactions with His41 (Fig 5D). Furthermore, the most important difference between rep1 and rep2 was the absence of a critical hydrogen bond at Glu166, which caused the ligand to exit the protein from its other side (S4 Video).

Ultimately, in the last and also in the shortest replica with the time of 18 ns, due to lack of significant catalytic VdW forces (Fig 5F and 5J), Glu166 by forming hydrogen bond (Fig 5G), caused the ligand to be pulled out of the active site in the first state. In the second intermediate state (Fig 5N), Tyr126 and Ser139 residues by VdW (Fig 5K) and hydrogen bonding interactions with acetylamino phenyl kept the inhibitor in the protein exposure, out of the binding pocket for 14ns, respectively (S5 Video). In the results of experimental research, a long list of important residues is reported. Between these reported important residues, Met49, Glu166, and Gln189 correlate with our results [26].

In contrast with ML188, more water molecules in the native binding pose of this inhibitor caused essential interactions to become water-mediated from the first moment of simulation (Fig 6A and 6B). Also, by time passing, due to benzotriazole free fragment', more space was created for water molecules insertion into the binding pocket, and ligand got unbound more rapidly (Fig 6C).

We proceeded to simulate the unbinding mechanism of two additional compounds to confirm and complete the information obtained and perhaps even identify new key factors. The

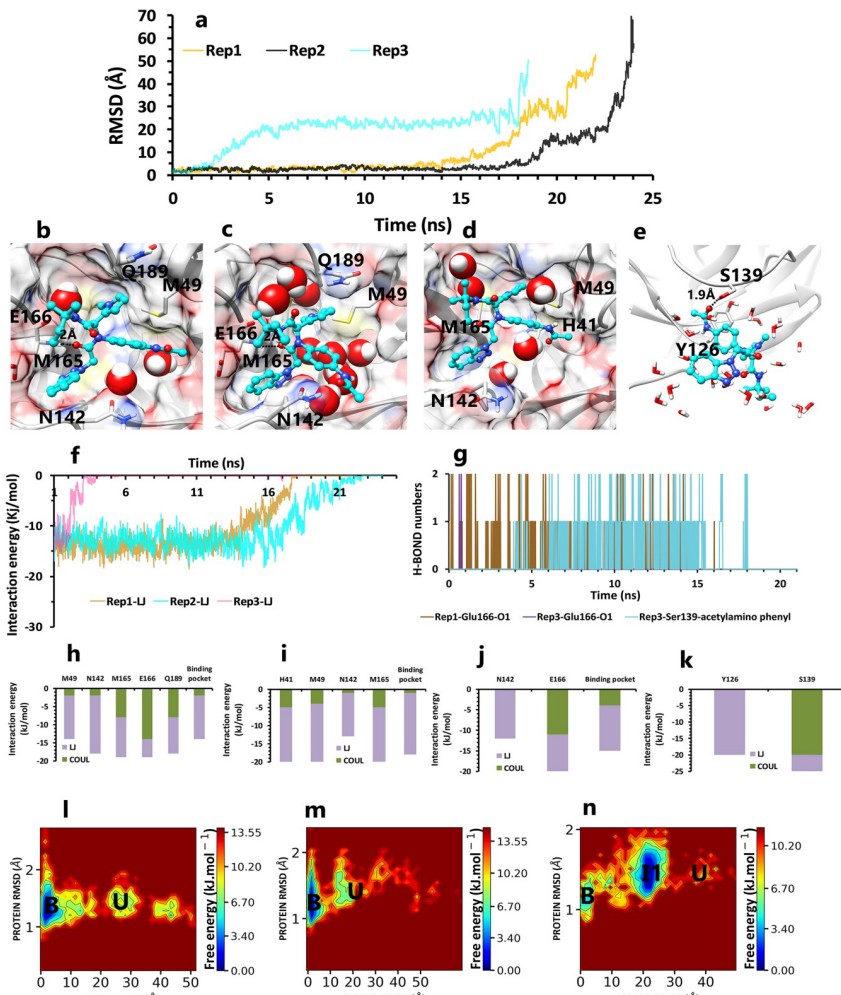

**Fig 5. The details of ML300 unbinding pathways in three replicas. A,** RMSD values of the ligand from binding pose to complete unbinding in three replicas. **B,** The interactions between particular fragments of ligand and important residues in the crystallographic binding pose of rep1 **C,** The new interaction between the acetylamino phenyl fragment and Asn142 after its rotation in rep1(frame 1207 in the trajectory file). **D,** The interactions between particular fragments of ligand and important residues in rep2 (frame 372 in the trajectory file). **E,** The last interactions of ligand and protein in the second intermediate state of rep3 (frame 1172 in the trajectory file). **F,** The average of most important residues Lennard-Jones (LJ) energies of the protein-ligand complex in rep1, 2, and 3 as a function of time. **G,** Hydrogen bond numbers of Glu166 with the oxygen atom close to benzotriazole fragment in rep1 and 3, and Ser139 with acetylamino phenyl fragment in rep3. **H and I,** The average of most important interaction energies of the protein-ligand complex in rep1 and 2, respectively. **J and K,** The average of most important interaction energies of the protein-ligand complex in the first and second intermediate states of rep3. **L, M** and **N,** The free energy landscape of rep1, 2, and 3 during the unbinding process (bound state (B), intermediate state (I), unbound (U)), respectively.

potent compound in PDB ID 6W63, with a broad spectrum of anti-viral activities, was chosen to achieve this goal. Furthermore, since this compound has some structural similarities to ML188, we were interested in understanding what was happening between this inhibitor and the protein while it was unbinding. In the first and second replicas with 55 ns and 61 ns, respectively (Fig 7A), the pyridine ring of the compound had cation–pi interaction with His163 in the deep part of the binding pocket (Fig 7B), and also the backbone of Gly143 had H-bond interaction with the oxygen atom of imidazole fragment in the shallow part of the pocket (Fig 7F). These potent interactions and VdW interaction between Met165 and butyl-phenyl fragment were the most important protein-inhibitor interactions in the native binding

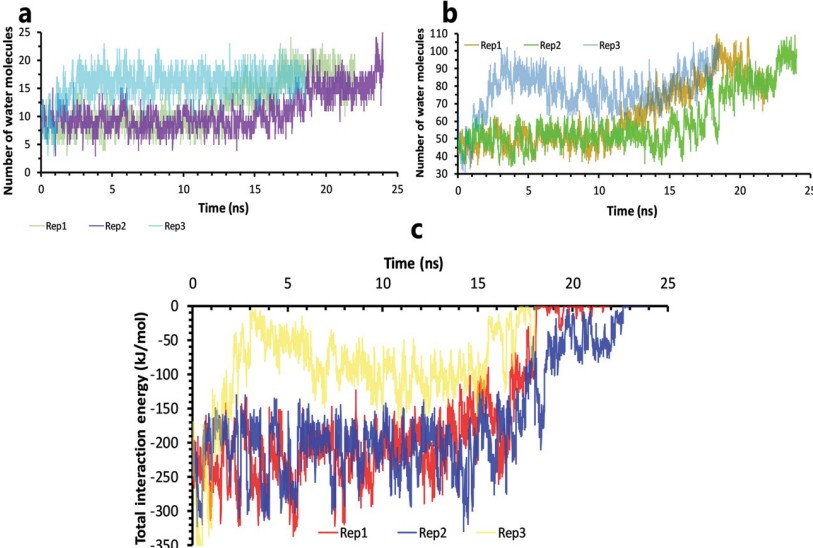

**Fig 6. The details of solvation effects on ML300 unbinding mechanisms. A**, Number of water molecules in the cut-off of 3.5 Å of the binding pocket residues, rep1, 2, and 3. **B,** Number of water molecules in the cut-off of 5 Å of the inhibitor, rep1, 2, and 3. **C,** The total interactions energies of protein-inhibitor complexes in rep1, 2, and 3.

pose (Fig 7H and 7I). Later, Asn142, by forming a H-bond with the oxygen atom of the imidazole fragment (Fig 7C and 7G), caused this fragment to be kept by two consecutive residues. This good binding pose was continued until 30ns, insofar as, Asn142 switched its H-bond to an imidazole ring (Fig 7D), and while Met 165 was still in interaction with the butylphenyl fragment, His163 lost its effect. So the ligand gradually moved from the deep part of the binding pocket to the surface area. Then, after time passing, by rotation of butylphenyl fragment, cyclohexylacetamide fragment formed H-bond with Glu166 (Fig 7E), and finally, the inhibitor unbound from Glu166 side.

In the final replica, due to the lack of a continuous binding pose key hydrogen bonds (Fig 7F and 7K), the ligand left the protein after 30 ns (Fig 7A). With more details, the lifetime of H-bond interaction between Gly143 and butylphenyl fragment was too short (95 ps), so this fragment could not be fixed and exposed to Met165. Like the last two replicas, butylphenyl fragment rotation and complete unbinding were observed sooner (Fig 7J). This inhibitor had only one state (Fig 7K–7M), and after weakening the binding pose interactions, the ligand did not trap in a serious state (S6 Video).

At the beginning of the unbinding process of the inhibitor3 in PDB ID 6W63, the crystalline water molecules are present at the native binding pose. In the following, more water molecules came into the binding site and broke important interactions (Fig 8A and 8B). Ultimately the ligand is unbound with complete solvation (Fig 8C).

Finally, the inhibitor in PDB ID 5RF7 was the last candidate because of its different structure compared with ML188. In this regard, all its replicas with times of 12, 16, and 17 ns, consider as a rapid unbinding pathway (Fig 9A). So in the binding pose of this pathway, while the 1,4-dimethylpiperazine fragment had only pi-sulfur interaction with Met165, the pyrrolo[2,3-b] pyridine fragment of inhibitor was in VdW and amino-pi interactions with Asn142, VdW interaction with Glu166, and had polar–pi interaction with Ser144 and also had Cation–pi interaction with His163 (Fig 9E–9G). Pyrrolo [2,3-b] pyridine, unlike 1,4-dimethylpiperazine fragment, was well kept by various interactions with both superficial and deep residues (Fig 9B). In the following, by rotation of methylpiperazine fragment

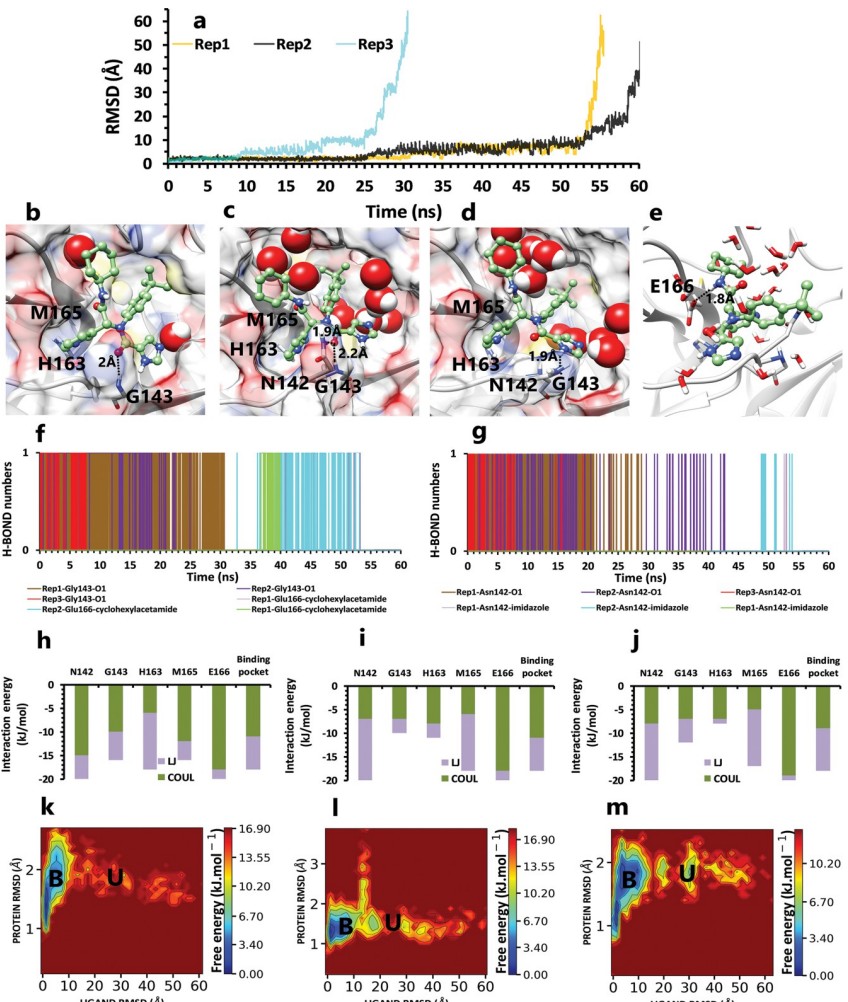

**Fig 7. The details of the inhibitor3 in PDB ID 6W63 unbinding pathways in three replicas. A,** RMSD values of the ligand from binding pose to complete unbinding in three replicas. **B,** The interactions between particular fragments of ligand and important residues in the crystallographic binding pose of rep1, 2, and 3. **C,** The interaction between the imidazole fragment and Asn142 in rep1, 2, and 3 (frame 607 in the trajectory file of rep1). **D,** The new interaction between Asn142 and imidazole ring in rep1, 2, and 3 (frame 2843 in the trajectory file of rep1). **E,** The new interaction between Glu166 and cyclohexylacetamide fragment, after butylphenyl fragment rotation in rep1, 2, and 3 (frame 4291 in the trajectory file of rep1). **F,** Hydrogen bond numbers of Gly143 and O1 and also Hydrogen bond numbers of Glu166 with cyclohexylacetamide in all replicas **G,** Hydrogen bond numbers of Asn142 and O1, and also imidazole ring in all replicas. **H, I,** and **J,** The average of most important interaction energies of the protein-ligand complex in rep1, 2, and 3, respectively. **K, L,** and **M** The free energy landscape of rep1, 2, and 3 during the unbinding process (bound state (B), intermediate state (I), unbound (U)), respectively.

toward Glu166, His163 loosed its strategic effect to let the ligand enter to the second intermediate state of the two longer replicas (rep1 and rep3) and be tapped in anion-pi interaction of Glu166 with pyrrolo[2,3-b] pyridine fragment (Fig 9C) as the last protein-inhibitor interaction (S7 Video). The second intermediate state of rep1 and 3 was unstable, as was the first state of all replicas (Fig 9H–9J). Even though all these inhibitor fragments were in serious interactions and there were no water molecules between these fragments and specific residues in the binding pose, there was enough space for water molecules to enter (Fig 9K and 9L). So this tiny ligand, by water-mediated interactions, loosed all its important interactions and unbound in a quick time (Fig 9D).

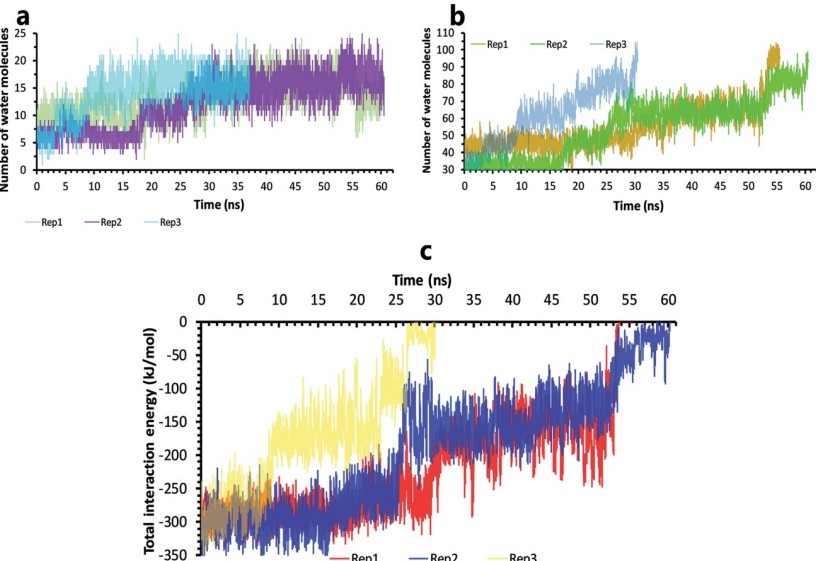

**Fig 8. The details of solvation effects on inhibitor3 in PDB ID 6W63 unbinding mechanisms. A**, Number of water molecules in the cut-off of 3.5 Å of the binding pocket residues, rep1, 2, and 3. **B,** Number of water molecules in the cut-off of 5 Å of the inhibitor, rep1, 2, and 3. **C,** The total interactions energies of protein-inhibitor complexes in rep1, 2, and 3.

## Conclusion

By putting together the atomic details of the unbinding pathways of selected inhibitors except inhibitor 4 in PDB ID 5RF7, one significant common weakness point was observed at various times in all replicas. There were no serious interactions between all fragments of inhibitors at specific residues in the binding pocket, so this factor was sufficient to weaken critical interactions. Almost free fragments could compete with other fragments for interactions with key residues. Even if they did not engage in competitive interactions, they could still change the inhibitors' positions and move closer to the exit path by rotating them. So the inhibitor in PDB ID 6W63 made serious noncovalent interactions with N142, G143, and H163 in the binding pocket, but due to its free 1-(1H-imidazol-4-yl) ethanone fragment could not stay longer in the binding pose.

In another study, hydrogen bonding interaction between inhibitor in PDB ID 3V3M and His163 was considered an essential interaction, but in our three replicas, the role of His163 in unbinding was not critical [14]. Based on the other research for the inhibitor in PDB ID 4MDS, Met49 and Gln189 were important residues correlated with our important residue list in keeping the inhibitor at the binding site [26].

Furthermore, water molecules played a functional role in all unbinding mechanisms by interfering and breaking important protein-inhibitor interactions. As time progressed, all inhibitors could not interact with all S regions of the binding pocket, and there was enough space for more water molecules to be inserted from outside. So the inhibitor 4 in PDB ID 5RF7, which did not have a free fragment, was too tiny and, in comparison with other inhibitors, occupied less space of binding pocket. So there was more space for water molecules to enter and caused less time to unbind due to the solvation effect. On the other hand, when the inhibitor occupies all space of the binding pocket, water molecules cannot penetrate under the ligand, and important inhibitor-protein interactions do not become water-mediated. In conclusion, the next series of noncovalent inhibitors should be designed to occupy all S regions of

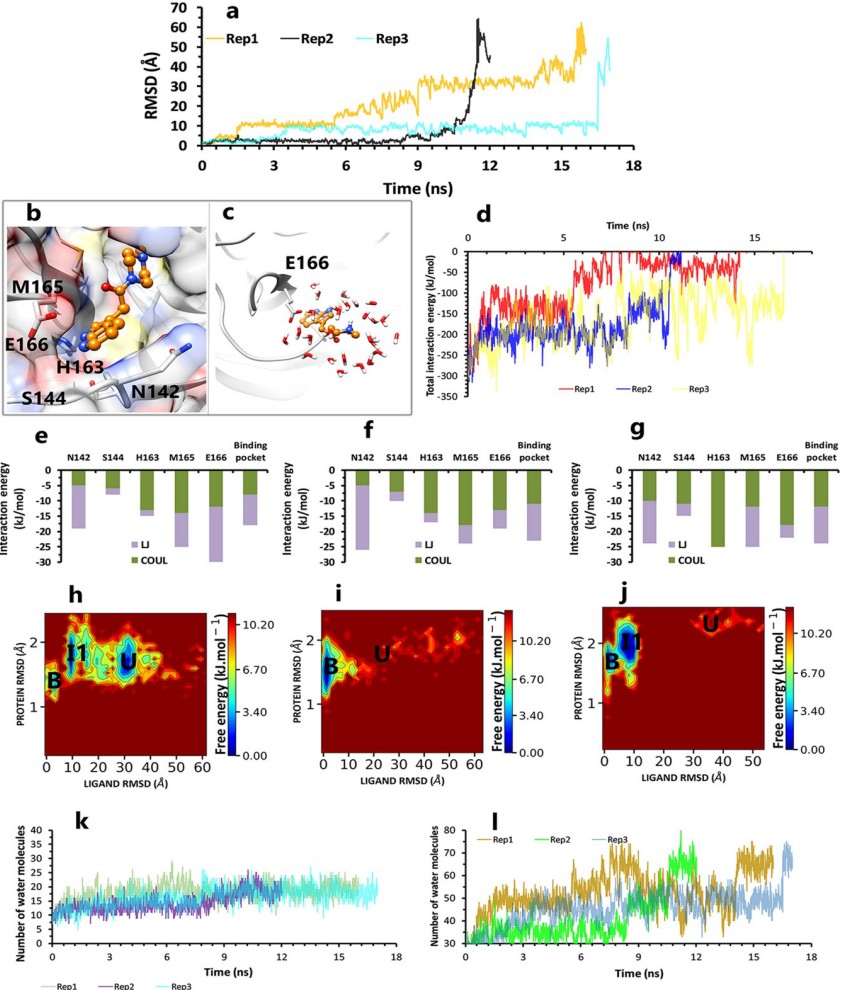

**Fig 9. The details of the inhibitor4 in PDB ID 5RF7 unbinding pathways in three replicas. A,** RMSD values of the ligand from binding pose to complete unbinding in three replicas. **B,** The interactions between particular fragments of ligand and important residues in the crystallographic binding pose of rep1, 2 and 3. **C,** The new interaction between the pyrrolo[2,3-b] pyridine fragment and Glu166 in rep1, 2, and 3 (frame 548 in the trajectory file of rep1). **D,** The total interaction energies of protein-inhibitor complexes in rep1, 2, and 3. **E, F,** and **G** The average of most important interaction energies of the protein-ligand complex in rep1, 2, and 3, respectively. **H, I,** and **J,** The free energy landscape of rep1, 2, and 3 during the unbinding process (bound state (B), intermediate state (I), unbound (U)), respectively. **K,** Number of water molecules in the cut-off of 3.5 Å of the binding pocket residues, rep1, 2, and 3. **L,** Number of water molecules in the cut-off of 5 Å of the inhibitor, rep1, 2, and 3.

the binding pocket to make maximum noncovalent interactions. This information is valuable for designing a new generation of inhibitors against this molecular target by fixing the weaknesses mentioned.

## Supporting information

**S1 Video. Video1, unbinding pathway of ML188 in rep1.**
(MP4)

**S2 Video. Video2, unbinding pathway of ML188 in rep2 and 3.**
(MP4)

**S3 Video. Video3, unbinding pathway of ML300 in rep1.**
(MP4)

**S4 Video. Video4, unbinding pathway of ML300 in rep2.**
(MP4)

**S5 Video. Video5, unbinding pathway of ML300 in rep3.**
(MP4)

**S6 Video. Video6, unbinding pathway of inhibitor 3 in PDB ID 6W63 in rep1-3.**
(MP4)

**S7 Video. Video7, unbinding pathway of inhibitor 4 in PDB ID 5RF7 in rep1-3.**
(MP4)

## Author Contributions

**Conceptualization:** Hassan Aryapour.

**Data curation:** Fereshteh Noroozi Tiyoula.

**Formal analysis:** Fereshteh Noroozi Tiyoula.

**Investigation:** Fereshteh Noroozi Tiyoula, Hassan Aryapour.

**Methodology:** Fereshteh Noroozi Tiyoula, Hassan Aryapour.

**Project administration:** Hassan Aryapour.

**Software:** Fereshteh Noroozi Tiyoula, Hassan Aryapour.

**Supervision:** Hassan Aryapour.

**Validation:** Fereshteh Noroozi Tiyoula, Hassan Aryapour.

**Visualization:** Fereshteh Noroozi Tiyoula.

**Writing – original draft:** Fereshteh Noroozi Tiyoula.

**Writing – review & editing:** Fereshteh Noroozi Tiyoula, Hassan Aryapour.

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
