## [Decision Letter · Decision Letter 0]

9 Nov 2021

PONE-D-21-32120Reconstruction of the unbinding pathways of noncovalent SARS-COV and SARS-COV-2 3CLpro inhibitors using Unbiased Molecular DynamicsPLOS ONE

Dear Dr. Aryapour,

Thank you for submitting your manuscript to PLOS ONE. After careful consideration, we feel that it has merit but does not fully meet PLOS ONE’s publication criteria as it currently stands. Therefore, we invite you to submit a revised version of the manuscript that addresses the points raised during the review process. Please submit your revised manuscript by Dec 24 2021 11:59PM. If you will need more time than this to complete your revisions, please reply to this message or contact the journal office at plosone@plos.org. Please include the following items when submitting your revised manuscript:A rebuttal letter that responds to each point raised by the academic editor and reviewer(s). You should upload this letter as a separate file labeled 'Response to Reviewers'.A marked-up copy of your manuscript that highlights changes made to the original version. You should upload this as a separate file labeled 'Revised Manuscript with Track Changes'.An unmarked version of your revised paper without tracked changes. You should upload this as a separate file labeled 'Manuscript'.

We look forward to receiving your revised manuscript.

Kind regards,

Chandrabose Selvaraj, Ph.D.

Academic Editor

PLOS ONE

Journal Requirements:

"YES

grant number 99-213-1 from Golestan University, Gorgan, Iran."

Reviewers' comments:

Reviewer's Responses to Questions

5. Review Comments to the Author

Reviewer #1: In the article entitled “Reconstruction of the unbinding pathways of noncovalent SARS-COV and SARSCOV-2 3CLpro inhibitors using Unbiased Molecular Dynamics” the authors performed an in silico analysis of the unbinding process for two SARS-CoV and two SARS-CoV-2 3CLpro inhibitors. While the idea of the research seems good and very useful and the research seems to be performed correctly, the discussion is lacking; there are several issues that require attention and need to be corrected and more thoroughly explained before the article is accepted:

1. Capitalization of abbreviations “SARS-CoV” and “SARS-CoV-2” needs to be corrected in the title, as well as in the text. Additionally, capitalization of other words, such as “viruses” in the abstract needs to be corrected.

2. Also, there seems to be an issue with font type and size which is inconsistent.

3. Throughout the manuscript, English should be corrected e.g. the sentence (lines 35-36): “Coronaviruses (CoVs), which have the largest genome among RNA viruses, are enveloped by single-stranded RNA viruses.” does not make any sense.

4. The sentence (lines 71-72): “On the other hand, investigation of the unbinding pathways of particular inhibitors, protein-ligand key interactions, and ligand's effectiveness or weakness can be obtained.” needs to be rephrased, it does not make a lot of sense.

5. The sentence (line 73): “Ultimately, the obtained results will be presented as a fully atomistic scenario [17].” needs to be rephrased (and font changed).

6. The sentence (lines 86-90): “Since, half-maximal inhibitory concentration (IC50) of SARS-COV 3CLpro inhibitors are measured experimentally before, and the 3CLpro of SARS-COV and SARS-CoV-2 are approximately the same, our research team for participating in the global solidarity trial decided to examine some noncovalent SARS-CoV and SARS-CoV-2 protease inhibitors' unbinding pathways by the SuMD, Comparatively.” is very confusing and needs to be rephrased. Also, what does “approximately the same” even mean? Additional clarification is needed and does this influence the binding site?

7. There is no discussion and comparison of the obtained results with results of other papers, have other researchers performed similar studies, are the conclusions the same, is there any difference in results or a new insight?

8. What are the differences between the SARS-CoV and SARS-CoV-2 3CLpro binding sites? How does this reflect on binding of the studied inhibitors and their selectivity? If there are not any differences, it should also be said.

9. In figures, such as Figure 3 l, m and n, should the word “state (S)” be “bound state”?

10. Figures in general are of low quality, I don’t know if this is a consequence of converting to the PDF format or not.

11. Can the authors explain the “free energy” term in e.g. Figure 2 m, n, o, is it the ∆GBIND or something else?

12. Can the authors also explain how come in some cases (e.g. Figure 3.c) the total interaction energy goes as low as -350 kJ/mol, what does this interaction energy represent?

13. In line 255, what does the expression “the almost deep part” mean?

14. In the Conclusion, English is especially poor: the sentence (lines 336-338): “All fragments of inhibitors did not interact seriously at specific residues in the binding pocket, so this factor was sufficient to weaken critical interactions, resulting in rapid unbinding.” is very confusing and has no meaning.

15. The same goes for the sentence after that (lines 338-341): “These almost free fragments could compete with other fragments for interactions with essential residues. Even if they did not enter into competitive interactions, they could change inhibitors' position and brought the inhibitor closer to the exit path by their rotations.” which needs to be rephrased.

16. The sentence (lines 353-355): “In conclusion, the next series of noncovalent inhibitors should be designed so that they can occupy all S regions of the binding pocket to make maximum noncovalent interactions.” Does not say anything new, this is common knowledge – the inhibitors should fill out the binding site as much as possible. Are there any concrete suggestions the authors have for development of future 3CLpro inhibitors, which interactions are good and should stay, which interactions are bad and should be corrected etc.?

Reviewer #2: The article "Reconstruction of the unbinding pathways of noncovalent SARS-COV and SARS-COV-2 3CLpro inhibitors using Unbiased Molecular Dynamics" is an interesting paper in which pure classical Molecular Dynamics Simulations have been employed to investigate the unbinding process pathways of two noncovalent inhibitors of SARS-COV and SARS-COV-2. Authors have used the PDB IDs: 3V3M, 4MDS, 6W63 and 5RF7 and they have explored the inhibitors comparing the protein-inhibitor interaction by means of unbiased molecular dynamics (UMD) simulations. Authors have found that when the inhibitor occupies all space of the binding pocket, water molecules cannot penetrate under the ligand, and important inhibitor-protein interactions do not become water-mediated. In other words, the inhibitor must occupy all S region to develop a proper binding, otherwise, water molecules can occupy this space and important interactions can get lost.

Comments:

1) The methodology of this work is correct, the analysis is well supported, and the results are partially interesting.

2) The SuMD MD simulation is interesting to unbind the inhibitor from the active site because the center of masses distance is increased in each window. Perhaps, for pure classical Molecular Dynamic Simulations, just 500 ps could be not enough to equilibrate the system. Have the authors tried to increase the simulation time by window to avoid possible equilibration problems?

3) The structure of the paper and the results are very clear. There are four systems: two inhibitors and two proteins (3CL of SAR-CoV and SARS-CoV-2). For each system, three replicas unbinding the inhibitor (from 25 to 65 ns). For the results, same figures and same analysis to compare the results properly. My major criticism about this paper is that the main goals from the results: on one hand, when the inhibitor is unbinding, water molecules are coming in the active site, so, when the inhibitor occupy the S region, the binding should be better because less water molecules are affecting the inhibitor-enzyme interaction, so the main idea is that “water molecules played a functional role in all unbinding mechanisms by interfering and breaking important protein-inhibitor interactions”, which is something very well known in from many other works On the other hand, the inhibitor interacts such as independent fragments with different parts of the proteins. I think these findings are not very novel, however, some results are very interesting such as Figures 2, 4 and 6, parts m, n and o. These maps should be explained by the authors more carefully.

4) The bibliography is insufficient. There are many QM/MM Molecular Dynamic simulations and classical simulations of many inhibitors of the 3CL enzyme of SAR-CoV and SARS-CoV-2 which must be included by the authors.

5) As minor points:

1) The structure of the inhibitors should be provided in the introduction section. In addition, the structure of the active site of the complex protein-inhibitor highlighting the critical distances should be showed as well.

2) Page 9, line 178 “j and k” looks like a typo mistake

3) Footnote of Figure 3. Cut-offs are 0.35 and 0.5 angstroms? I think there is a mistake is those data.

Reviewer #3: The manuscript ID PONE-D-21-32120 entitled “Reconstruction of the unbinding pathways of noncovalent SARS-COV and SARS-COV-2 3CLpro inhibitors using Unbiased Molecular Dynamics” is quite a good study. The main protease (3CLpro) is one of the essential components of the SARS-COVs viral life cycle, which makes it an interesting target for overpowering Viruses. Although many covalent and noncovalent inhibitors have been designed to inhibit this molecular target, none have gained FDA approval as a drug. Because of the high rate of COVID-19 pandemic development, in addition to laboratory research, we require in silico methods to accelerate rational drug design. The unbinding pathways of two SARS-COV and SARS-COV-2 noncovalent inhibitors with the PDB IDs: 3V3M, 4MDS, 6W63, 5RF7 were explored from a comparative perspective using the unbiased molecular dynamics (UMD) simulations. We uncovered common weak points for selected inhibitors that could not have significant interactions with a binding pocket at specific residues by all their fragments. So water molecules entered the free binding S regions and weakened protein-inhibitor key interactions gradually. Finally, the authors believed that these results will help to design new potent inhibitors against SARS-CoV-2.

I appreciate the authors for their great effort of enclosing all the supplementary materials. The videos are really nice.

However, the following queries should be addressed before submitting a revision

1) In the abstract, the author provided more general info instead of the results of the study. In addition, the conclusion and methods are not appropriate---make it clear

2) In the introduction section, the author might start with SARS first, then SARS-2 to follow the uniformity of the title of the study.

3) Page 8 – lines 46-49: “The domains I and II have 46 antiparallel β barrel structures responsible for the catalytic reaction, while domain III has α-helices (Fig 1a, b). Since the dimeric form of Mpro is catalytically active, intermolecular interactions between the helical domains play an essential role in activating the enzyme [8, 9]. ------ It is not clear. What is the role of domain III?

4) What is unbiased molecular dynamics (UMD) and its advantages? Brief up 1 or 2 lines.

5) This line seems to be meaningless – “MD simulation has been a standard method of predicting hidden information in many science fields since the 1950s”----what is meant by hidden info by MD simulations?

6) What is the correlation of the above lines and “in judging candidate drugs, the bioavailability, selectivity, metabolic properties, and binding affinity of the designed inhibitor to its target protein are all important? [16].”

7) It is not an appropriate word “ligand's effectiveness or weakness can be obtained”. It may be ligands interacting efficiently with the target

8) These lines are a bit confusing or meaningless “Since, half-maximal inhibitory concentration (IC50) of SARS-COV 3CLpro inhibitors are measured experimentally before, and the 3CLpro of SARS-COV and SARS CoV-2 are approximately the same, our research team for participating in the global solidarity trial decided to examine some noncovalent SARS-CoV and SARS-CoV-2 protease inhibitors' unbinding pathways by the SuMD, Comparatively.” From MD authors defining IC50 value?

9) This is unclear: we performed 12 simulations, three replicas for each complex, by the SuMD method with some modifications---what are 12 simulations? Whether author’s starts from 1ps to 500ps or 1ps to 1500 ps defined as replicas? Because the following line authors mentioned, “------to extend the next 500 ps simulation”.

10) Reference missing for UCSF Chimera

11) Author may improve the discussion because in the results and discussion section authors described only results.

12) How do the authors create a free energy plot? Which package?

13) In all the figures, authors may reduce the thickness of the lines to get better visualization---figures are not in good resolution

14) Conclusion should be improved.

15) The language of the manuscript should meet the journal’s adequate standard

---

## [Author Response · Author response to Decision Letter 0]

11 Dec 2021

Dear Prof. Chandrabose Selvaraj;

I am pleased to mail you the revised article titled "Reconstruction of the unbinding pathways of noncovalent SARS-CoV and SARS-CoV-2 3CLpro inhibitors using Unbiased Molecular Dynamics". All comments have been answered upon reviewers' recommendations, and the corrections were made in manuscript #PONE-D-21-32120. It is our pleasure to hear your feedback and any other suggestions relevant to the paper.

Sincerely,

Hassan Aryapour

Reviewer #1:

Comment #1:

Capitalization of abbreviations "SARS-CoV" and "SARS-CoV-2" needs to be corrected in the title, as well as in the text. Additionally, capitalization of other words, such as "viruses" in the abstract needs to be corrected.

Answer #1:

The requested items were corrected.

Comment #2:

Also, there seems to be an issue with font type and size which is inconsistent.

Answer #2:

The entire text was reviewed and corrected.

Comment #3:

Throughout the manuscript, English should be corrected e.g. the sentence (lines 35-36): "Coronaviruses (CoVs), which have the largest genome among RNA viruses, are enveloped by single-stranded RNA viruses." does not make any sense.

Answer #3:

A new version of this sentence is highlighted in (lines 40-42).

Comment #4:

The sentence (lines 71-72): "On the other hand, investigation of the unbinding pathways of particular inhibitors, protein-ligand key interactions, and ligand's effectiveness or weakness can be obtained." needs to be rephrased, it does not make a lot of sense.

Answer #4:

A new version of this sentence is highlighted in (lines 85-87).

Comment #5:

The sentence (line 73): "Ultimately, the obtained results will be presented as a fully atomistic scenario [17]." needs to be rephrased (and font changed).

Answer #5:

A new version of this sentence is highlighted in (lines 88-89).

Comment #6:

The sentence (lines 86-90): "Since, half-maximal inhibitory concentration (IC50) of SARS-COV 3CLpro inhibitors are measured experimentally before, and the 3CLpro of SARS-COV and SARS-CoV-2 are approximately the same, our research team for participating in the global solidarity trial decided to examine some noncovalent SARS-CoV and SARS-CoV-2 protease inhibitors' unbinding pathways by the SuMD, Comparatively." is very confusing and needs to be rephrased. Also, what does "approximately the same" even mean? Additional clarification is needed and does this influence the binding site?

Answer #6:

This paragraph was rewritten for better understanding. We used "approximately the same", based on multiple sequence alignment, which reported 96.08% identity and 98.7% similarity between 3CLpro of SARS-CoV and SARS-CoV-2. There is no difference for the 3CLpro binding sites of these two viruses, so we added this information to the new version of this paragraph.

Comment #7:

There is no discussion and comparison of the obtained results with results of other papers, have other researchers performed similar studies, are the conclusions the same, is there any difference in results or a new insight?

Answer #7:

PDB IDs 3V3M and 4MDS have other research, and we added a comparison of their results with ours in the result and conclusion sections in the (lines 187-190 & 240-242 & 371-375) but for the two other PDB IDs (6W63 and 5RF7) there are no articles.

Comment #8:

What are the differences between the SARS-CoV and SARS-CoV-2 3CLpro binding sites? How does this reflect on binding of the studied inhibitors and their selectivity? If there are not any differences, it should also be said.

Answer #8:

The comment was answered alongside Comment #6.

Comment #9:

In figures, such as Figure 3 l, m and n, should the word "state (S)" be "bound state"?

Answer #9:

In all figures, state words were replaced with bound states (B).

Comment #10:

Figures in general are of low quality, I don't know if this is a consequence of converting to the PDF format or not.

Answer #10:

We fixed the problem of low-quality figures caused by PDF conversion.

Comment #11:

Can the authors explain the "free energy" term in e.g. Figure 2 m, n, o, is it the ∆GBIND or something else?

Answer #11:

 The free energy landscape plots were obtained using "gmx sham" as we mentioned in the figures. Actually, "gmx sham" leads to plot Gibbs's free energy landscapes. 

Comment #12:

Can the authors also explain how come in some cases (e.g. Figure 3.c) the total interaction energy goes as low as -350 kJ/mol, what does this interaction energy represent?

Answer #12:

Total interaction energies plots are obtained from the sum of the coulomb and Lennard-jones contribution values. We have maximum interaction energies in the bound state of the protein-ligand complex. As time passes and protein-ligand interactions weaken, the values of the plot increase and eventually reach zero until full unbinding. So all the interactions energies plots' trend is correlated with the unbinding process from the bound state to the unbound state. Also, -350 kJ / mol values in some plots and -300 kJ / mol values in others indicate the value of bound state interaction energies. Since these interactions measure in vacuum conditions, these values cannot prove that the inhibitor with a lower interaction energy value in a bound state is more potent. These plots should be examined along with the average of the most important interaction energies for better understanding. 

Comment #13:

In line 255, what does the expression "the almost deep part" mean?

Answer #13:

The word "almost" is deleted.

Comment #14:

In the Conclusion, English is especially poor: the sentence (lines 336-338): "All fragments of inhibitors did not interact seriously at specific residues in the binding pocket, so this factor was sufficient to weaken critical interactions, resulting in rapid unbinding." is very confusing and has no meaning.

Answer #14:

A new version of this sentence is highlighted in (lines 358-361).

Comment #15:

The same goes for the sentence after that (lines 338-341): "These almost free fragments could compete with other fragments for interactions with essential residues. Even if they did not enter into competitive interactions, they could change inhibitors' position and brought the inhibitor closer to the exit path by their rotations." which needs to be rephrased.

Answer #15:

A new version of this sentence is highlighted in (lines 361-365).

Comment #16:

The sentence (lines 353-355): "In conclusion, the next series of noncovalent inhibitors should be designed so that they can occupy all S regions of the binding pocket to make maximum noncovalent interactions." Does not say anything new, this is common knowledge – the inhibitors should fill out the binding site as much as possible. Are there any concrete suggestions the authors have for development of future 3CLpro inhibitors, which interactions are good and should stay, which interactions are bad and should be corrected etc.?

Answer #16:

We explained everything, even common knowledge, because we wanted to report all atomistic details. Moreover, we explained all favorable and unfavorable interactions in the results and discussion sections for all inhibitors' unbinding processes. The most potent interactions are also listed in the conclusion section (lines 367-370).

Reviewer #2:

Comment #1:

The methodology of this work is correct, the analysis is well supported, and the results are partially interesting.

Comment #2:

The SuMD MD simulation is interesting to unbind the inhibitor from the active site because the center of masses distance is increased in each window. Perhaps, for pure classical Molecular Dynamic Simulations, just 500 ps could be not enough to equilibrate the system. Have the authors tried to increase the simulation time by window to avoid possible equilibration problems?

Answer #2:

Equilibration was applied by performing 100 ps of NVT and 300 of NPT. The 500 ps simulation is not for equilibration. The method section has been improved for better understanding.

Comment #3:

The structure of the paper and the results are very clear. There are four systems: two inhibitors and two proteins (3CL of SAR-CoV and SARS-CoV-2). For each system, three replicas unbinding the inhibitor (from 25 to 65 ns). For the results, same figures and same analysis to compare the results properly. My major criticism about this paper is that the main goals from the results: on one hand, when the inhibitor is unbinding, water molecules are coming in the active site, so, when the inhibitor occupy the S region, the binding should be better because less water molecules are affecting the inhibitor-enzyme interaction, so the main idea is that "water molecules played a functional role in all unbinding mechanisms by interfering and breaking important protein-inhibitor interactions", which is something very well known in from many other works On the other hand, the inhibitor interacts such as independent fragments with different parts of the proteins. I think these findings are not very novel, however, some results are very interesting such as Figures 2, 4 and 6, parts m, n and o. These maps should be explained by the authors more carefully.

Answer #3:

For the "On the other hand, the inhibitor interacts such as independent fragments with different parts of the proteins. I think these findings are not very novel" we explained in the answer of Comment #16 of reviewe1. Also, we added a more detailed explanation of the Free energy landscape plots in the method section.

Comment #4:

The bibliography is insufficient. There are many QM/MM Molecular Dynamic simulations and classical simulations of many inhibitors of the 3CL enzyme of SAR-CoV and SARS-CoV-2 which must be included by the authors.

Answer #4:

The QM/MM Molecular Dynamic simulations are a hybrid method that is entirely unrelated to our work. Our study is MM Molecular Dynamic simulations, which there is no related project to mention.

Comment #5-1:

The structure of the inhibitors should be provided in the introduction section. In addition, the structure of the active site of the complex protein-inhibitor highlighting the critical distances should be showed as well.

Answer #5-1:

All fragments of each inhibitor are existence in the figures of the result section. If we change the location of these pictures, the readership of this article may get board to switch to the introduction section and again back to the result section. The active site domain of the protein is explained in Figure1, and a binding pocket of the complex protein-inhibitor is picturizing separately in Figures 2, 4, 6, and 8.

Comment #5-2:

Page 9, line 178 "j and k" looks like a typo mistake

Answer #5-2:

This line was rechecked.

Comment #5-3:

Footnote of Figure 3. Cut-offs are 0.35 and 0.5 angstroms? I think there is a mistake is those data.

Answer #5-3:

True values of Cut-offs were replaced in figures' captions.

Reviewer #3:

Comment #1:

In the abstract, the author provided more general info instead of the results of the study. In addition, the conclusion and methods are not appropriate---make it clear.

Answer #1:

More information about the conclusion is added in the abstract section. Also, the Conclusion and method sections were improved.

Comment #2:

In the introduction section, the author might start with SARS first, then SARS-2 to follow the uniformity of the title of the study.

Answer #2:

The requested information was added in the introduction section (lines 31-32)

Comment #3:

Page 8 – lines 46-49: "The domains I and II have 46 antiparallel β barrel structures responsible for the catalytic reaction, while domain III has α-helices (Fig 1a, b). Since the dimeric form of Mpro is catalytically active, intermolecular interactions between the helical domains play an essential role in activating the enzyme [8, 9]. ------ It is not clear. What is the role of domain III?

Answer #3:

The requested information was added in the introduction section (line 53)

Comment #4:

What is unbiased molecular dynamics (UMD) and its advantages? Brief up 1 or 2 lines.

Answer #4:

The requested information was added in the introduction section (lines 75-77)

Comment #5:

This line seems to be meaningless – "MD simulation has been a standard method of predicting hidden information in many science fields since the 1950s"----what is meant by hidden info by MD simulations?

Answer #5:

Since the experimental research gives information based on x-ray crystallographic binding pose, results are statics and limited, but MD simulations are dynamic and give information about what is happening during unbinding pathways. Therefore, this information cannot be obtained through experimental research and is regarded as hidden.

Comment #6:

What is the correlation of the above lines and "in judging candidate drugs, the bioavailability, selectivity, metabolic properties, and binding affinity of the designed inhibitor to its target protein are all important? [16].”

Answer #6:

For better understanding, we have rewritten this sentence (lines 81-84).

Comment #7:

It is not an appropriate word "ligand's effectiveness or weakness can be obtained". It may be ligands interacting efficiently with the target

Answer #7:

 The "ligands interacting efficiently with the target" was replaced with "ligand's effectiveness or weakness can be obtained" in (lines 86-87).

Comment #8:

These lines are a bit confusing or meaningless "Since, half-maximal inhibitory concentration (IC50) of SARS-COV 3CLpro inhibitors are measured experimentally before, and the 3CLpro of SARS-COV and SARS CoV-2 are approximately the same, our research team for participating in the global solidarity trial decided to examine some noncovalent SARS-CoV and SARS-CoV-2 protease inhibitors' unbinding pathways by the SuMD, Comparatively." From MD authors defining IC50 value?

Answer #8:

Experimental research provided the IC50 value. The reference is added in (line 104).

Comment #9:

This is unclear: we performed 12 simulations, three replicas for each complex, by the SuMD method with some modifications---what are 12 simulations? Whether author's starts from 1ps to 500ps or 1ps to 1500 ps defined as replicas? Because the following line authors mentioned, "------to extend the next 500 ps simulation".

Answer #9:

We improved the method section. Actually, we ran all simulations with a time window of 500 ps. We mentioned in the method section, "After finishing each run, the frame with the longest distance between selected spots was selected automatically to extend the next 500 ps simulation". These processes were continued until complete unbind was obtained, which is equal to a distance of 50 Å between the mentioned spots", So each replica is a separate simulation from binding pose to complete unbinding. 

Comment #10:

Reference missing for UCSF Chimera

Answer #10:

The reference is available in (line 113).

Comment #11:

Author may improve the discussion because in the results and discussion section authors described only results.

Answer #11:

The conclusion section changed to improve.

Comment #12:

How do the authors create a free energy plot? Which package?

Answer #12:

The requested information was added in the method section (lines 144-150).

Comment #13:

In all the figures, authors may reduce the thickness of the lines to get better visualization---figures are not in good resolution.

Answer #13:

All figures were rebulided again, but for more precise images, we reduced the thickness of the lines, resulting in plots that faded

Comment #14:

Conclusion should be improved.

Answer #14:

The conclusion section changed to improve.

Comment #15:

The language of the manuscript should meet the journal's adequate standard.

Answer #15:

The requested work is done.

---

## [Decision Letter · Decision Letter 1]

21 Dec 2021

PONE-D-21-32120R1Reconstruction of the unbinding pathways of noncovalent SARS-CoV and SARS-CoV-2 3CLpro inhibitors using Unbiased Molecular DynamicsPLOS ONE

Dear Dr. Aryapour,

Thank you for submitting your manuscript to PLOS ONE. After careful consideration, we feel that it has merit but does not fully meet PLOS ONE’s publication criteria as it currently stands. Therefore, we invite you to submit a revised version of the manuscript that addresses the points raised during the review process.

ACADEMIC EDITOR: You may want to upload a letter of editing from a professional agency or native-English speaker in addition to the reviewer's comments because reviewers have expressed a concern about the language part.

We look forward to receiving your revised manuscript.

Kind regards,

Chandrabose Selvaraj, Ph.D.

Academic Editor

PLOS ONE

Journal Requirements:

Reviewers' comments:

Reviewer's Responses to Questions

6. Review Comments to the Author

Reviewer #1: The authors have significantly improved the manuscript and added further discussion of their results. There are still significant language issues that need to be corrected throughout the manuscript (with e.g. lines 39-40 „Coronaviruses (CoVs), an RNA virus with the largest genome, are enveloped by a single-stranded RNA virus.“ – coronaviruses are enveloped RNA viruses, they are not enveloped BY a single stranded RNA virus; and lines 99-101: “So our research team, for participating in the solidarity clinical trial for COVID-19 treatments, decided to examine some noncovalent SARS-CoV and SARS-CoV-2 protease inhibitors' unbinding pathways by the SuMD, Comparatively.“ which is still grammatically very incorrect). Additionally, my comment about the lack of discussion and comparison with results from other researchers is still standing. The authors have improved their discussion, but they have obtained a lot of new information which could have been more thoroughly discussed and compared, it would further increase the significance and value of their work. Nonetheless, even in the present state, I believe that this research should be accepted for publishing.

Since the PLOS ONE’s states “PLOS ONE does not copyedit accepted manuscripts, so the language in submitted articles must be clear, correct, and unambiguous. Any typographical or grammatical errors should be corrected at revision.”, I must ask the authors to find someone sufficiently proficient in English to proofread the article and correct all grammatical errors.

Reviewer #2: 1) Answer #5-1:

All fragments of each inhibitor are existence in the figures of the result section. If we change the location of these pictures, the readership of this article may get board to switch to the introduction section and again back to the result section. The active site domain of the protein is explained in Figure1, and a binding pocket of the complex protein-inhibitor is picturizing separately in Figures 2, 4, 6, and 8.

Comment#5-1:

I still think that a scheme of the inhibitors in the introduction section would increase the clarity of the study.

2) Answer #3:

For the "On the other hand, the inhibitor interacts such as independent fragments with different parts of the proteins. I think these findings are not very novel" we explained in the answer of Comment #16 of reviewe1. Also, we added a more detailed explanation of the Free energy landscape plots in the method section.

From the free energy landscapes, would be possible to predict a free energy of binding value of each system?

3) The initial Molecular Dynamic Simulations should be longer to obtain an equilibrated configuration of the proposed models. Before SuMDs, the system should be very well equilibrated. You can see many studies about 3CL enzyme in which classical MD of microsecond timescales is performed before to study binding or reaction processes. The initial protocol of this work does not allow, perhaps, to explore important configurations of the enzyme that should be considered in the study. Authors should explore this way carefully for future studies.

---

## [Author Response · Author response to Decision Letter 1]

9 Jan 2022

Dear Prof. Chandrabose Selvaraj;

I am pleased to mail you the revised article titled "Reconstruction of the unbinding pathways of noncovalent SARS-CoV and SARS-CoV-2 3CLpro inhibitors using Unbiased Molecular Dynamics". All comments have been answered upon reviewers' recommendations, and the corrections were made in manuscript #PONE-D-21-32120R1. It is our pleasure to hear your feedback and any other suggestions relevant to the paper.

Sincerely,

Hassan Aryapour

ACADEMIC EDITOR:

You may want to upload a letter of editing from a professional agency or native-English speaker in addition to the reviewer's comments because reviewers have expressed a concern about the language part.

Answer:

The entire document has been reviewed by someone who has a better understanding of English and checked for existing problems in the text

Journal Requirements:

Answer:

The references list was reviewed once more carefully, and no reference was changed.

Reviewer #1:

The authors have significantly improved the manuscript and added further discussion of their results. There are still significant language issues that need to be corrected throughout the manuscript (with e.g. lines 39-40 "Coronaviruses (CoVs), an RNA virus with the largest genome, are enveloped by a single-stranded RNA virus. "– coronaviruses are enveloped RNA viruses, they are not enveloped BY a single stranded RNA virus; and lines 99-101: "So our research team, for participating in the solidarity clinical trial for COVID-19 treatments, decided to examine some noncovalent SARS-CoV and SARS-CoV-2 protease inhibitors' unbinding pathways by the SuMD, Comparatively. "which is still grammatically very incorrect). Additionally, my comment about the lack of discussion and comparison with results from other researchers is still standing. The authors have improved their discussion, but they have obtained a lot of new information which could have been more thoroughly discussed and compared, it would further increase the significance and value of their work. Nonetheless, even in the present state, I believe that this research should be accepted for publishing.

Answer #2:

The lines 39 and 40 and also lines 99-101 were replaced with the corrected version in (lines 39-40 and 111-114).

For the comparison, our research method is different from the mentioned experimental research. They examined important protein-inhibitor interactions only in a binding pose, so we could compare only their ultimate result with a little part of our work.

Reviewer #2: 1) Answer #5-1:

Reviewer #2: 1) Answer #5-1:

All fragments of each inhibitor are existence in the figures of the result section. If we change the location of these pictures, the readership of this article may get board to switch to the introduction section and again back to the result section. The active site domain of the protein is explained in Figure1, and a binding pocket of the complex protein-inhibitor is picturizing separately in Figures 2, 4, 6, and 8.

Comment#5-1:

I still think that a scheme of the inhibitors in the introduction section would increase the clarity of the study.

Answer:

Inhibitors' structure image was replaced in the introduction section (Fig2 and lines 62-69).

2) Answer #3:

For the "On the other hand, the inhibitor interacts such as independent fragments with different parts of the proteins. I think these findings are not very novel" we explained in the answer of Comment #16 of reviewe1. Also, we added a more detailed explanation of the Free energy landscape plots in the method section.

From the free energy landscapes, would be possible to predict a free energy of binding value of each system?

Answer:

For calculating binding free energy, there are some ways involving: alchemical transfer method (ATM), linear interaction energy (LIE), PDLD/S-LRA method, alchemical free energy perturbation (FEP), BAR method, and so on. But, We computed the free energy landscape (FEL) to capture inhibitors' lowest energy stable states during unbinding by gmx sham module (https://manual.gromacs.org/documentation/2018/onlinehelp/gmx-sham.html). 

This module calculates free energy landscapes by computing the joint probability distribution from the two-dimensional plane constructed using two quantities (in our case, they were ligand RMSD (as the x-axis) and protein RMSD (as the y-axis)). Conformations sampled during the simulation were projected on this two-dimensional plane, and the number of points occupied by each cell was counted. The grid cell containing the maximum number of points is then assigned as the reference cell, with a free energy value of zero. Free energies for all the other cells were assigned with respect to this reference cell using the following equation:

ΔG(x,y) = -KbT ln P(x,y)/Pmin

P(x,y) estimates the probability density function obtained from a histogram of MD data, and Pmin is the maximum of the probability density function. Kb is the Boltzmann constant, and T is the temperature corresponding to each simulation. 

3): The initial Molecular Dynamic Simulations should be longer to obtain an equilibrated configuration of the proposed models. Before SuMDs, the system should be very well equilibrated. You can see many studies about 3CL enzyme in which classical MD of microsecond timescales is performed before to study binding or reaction processes. The initial protocol of this work does not allow, perhaps, to explore important configurations of the enzyme that should be considered in the study. Authors should explore this way carefully for future studies.

Answer:

Thanks for your valuable advice, and we will consider them for our future research.

---

## [Decision Letter · Decision Letter 2]

17 Jan 2022

Reconstruction of the unbinding pathways of noncovalent SARS-CoV and SARS-CoV-2 3CLpro inhibitors using Unbiased Molecular Dynamics

PONE-D-21-32120R2

Dear Dr. Aryapour,

We’re pleased to inform you that your manuscript has been judged scientifically suitable for publication and will be formally accepted for publication once it meets all outstanding technical requirements.

Kind regards,

Chandrabose Selvaraj, Ph.D.

Academic Editor

PLOS ONE

Additional Editor Comments (optional):

Reviewers' comments:

Reviewer's Responses to Questions

**Comments to the Author**

1. If the authors have adequately addressed your comments raised in a previous round of review and you feel that this manuscript is now acceptable for publication, you may indicate that here to bypass the “Comments to the Author” section, enter your conflict of interest statement in the “Confidential to Editor” section, and submit your "Accept" recommendation.

Reviewer #1: All comments have been addressed

4. Have the authors made all data underlying the findings in their manuscript fully available?

Reviewer #1: Yes

---

## [Editor Report · Acceptance letter]

28 Jan 2022

PONE-D-21-32120R2 

Reconstruction of the unbinding pathways of noncovalent SARS-CoV and SARS-CoV-2 3CLpro inhibitors using Unbiased Molecular Dynamics simulations 

Dear Dr. Aryapour:

I'm pleased to inform you that your manuscript has been deemed suitable for publication in PLOS ONE. Congratulations! Your manuscript is now with our production department. 

Kind regards, 

on behalf of

Dr. Chandrabose Selvaraj 

Academic Editor

PLOS ONE